# Tackling Dimensional Collapse toward Comprehensive Universal Domain Adaptation

**Hung-Chieh Fang** [1]  **Po-Yi Lu** [1]  **Hsuan-Tien Lin** [1]

## Abstract

Universal Domain Adaptation (UniDA) addresses unsupervised domain adaptation where target classes may differ arbitrarily from source ones, except for a shared subset. A widely used approach, partial domain matching (PDM), aligns only shared classes but struggles in extreme cases where many source classes are absent in the target domain, underperforming the most naive baseline that trains on only source data. In this work, we identify that the failure of PDM for extreme UniDA stems from dimensional collapse (DC) in target representations. To address target DC, we propose to jointly leverage the alignment and uniformity techniques in self-supervised learning on the unlabeled target data to preserve the intrinsic structure of the learned representations. Our experimental results confirm that SSL consistently advances PDM and delivers new state-of-the-art results across a broader benchmark of UniDA scenarios with different portions of shared classes, representing a crucial step toward truly comprehensive UniDA. Project page: https://dc-unida.github.io/

## 1. Introduction

While deep learning and machine learning have achieved remarkable success across a wide range of tasks, they still struggle when there is a distribution shift between the training and testing data. Unsupervised Domain Adaptation (UDA) (Pan & Yang, 2009) addresses this challenge by transferring knowledge from a labeled source domain with a known distribution to an unlabeled target domain that may follow a different distribution. Arguably the simplest UDA setting is closed-set UDA (Ganin et al., 2016; Long et al.,

[1]National Taiwan University. Correspondence to: Hung-Chieh Fang <b09902106@csie.ntu.edu.tw>, Hsuan-Tien Lin <htlin@csie.ntu.edu.tw>.

*Proceedings of the 42$^{st}$ International Conference on Machine Learning*, Vancouver, Canada. PMLR 267, 2025. Copyright 2025 by the author(s).

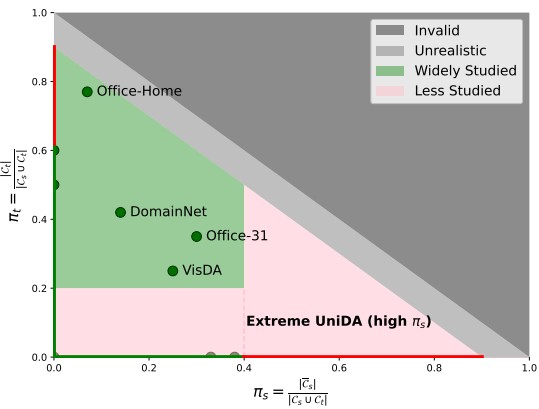

*Figure 1.* **Spectrum of UniDA**: The dark region indicates invalid cases with negative shared classes, while the light gray region represents unrealistic scenarios with few or no shared classes. The X-axis indicates Partial DA, and the Y-axis indicates Open-set DA.

2018; Jiang et al., 2020), which assumes that the label sets of the source domain ($\mathcal{C}_s$) and the target domain ($\mathcal{C}_t$) are identical ($\mathcal{C}_s = \mathcal{C}_t$).

More flexible setups such as Open-set Domain Adaptation ($\mathcal{C}_s \subset \mathcal{C}_t$) and Partial Domain Adaptation ($\mathcal{C}_s \supset \mathcal{C}_t$) have been studied (Panareda Busto & Gall, 2017; Cao et al., 2018), addressing scenarios where $\mathcal{C}_t$ has more or fewer classes than $\mathcal{C}_s$. Universal Domain Adaptation (UniDA) (You et al., 2019) further loosens the limit and unifies those setups by not assuming any containment relation between $\mathcal{C}_s$ and $\mathcal{C}_t$. Specifically, UniDA assumes unknown shared classes to overlap between $\mathcal{C}_s$ and $\mathcal{C}_t$, while allowing each set to contain its private classes. The setup is tasked with classifying target-domain examples as either one of the shared classes or as an *unseen* class over arbitrary divisions of source-private, shared, and target-private label sets.

Addressing the gap between two domains in UDA is commonly approached by combining a loss on the labeled source data with another loss on *domain matching*, which aims to align feature distributions between domains to enable effective knowledge transfer (Ganin et al., 2016; Courty et al., 2016; Tzeng et al., 2017; Zellinger et al., 2017; Jiang et al., 2020). However, in the UniDA setting, the presence of private classes in both domains, which are *unknown* dur-

ing training, poses a significant challenge for this approach. Matching feature distributions without separating shared and private classes can cause negative transfer, where knowledge transferred from private classes harms performance instead of improving it (Rosenstein et al., 2005).

To address this issue, Cao et al. (2018) introduced *partial domain matching* (PDM), which attempts to match only the data corresponding to the shared classes between domains. While an ideal PDM can effectively avoid negative transfer, it can be difficult to accurately determine and separate the examples that belong to the shared classes. Previous work has designed importance weight functions based on uncertainty measurement (You et al., 2019; Fu et al., 2020; Lifshitz & Wolf, 2021; Chen et al., 2022a; Chang et al., 2022) or relative distances in the embedding space (Saito et al., 2020; Li et al., 2021; Chen et al., 2022b; Lu et al., 2024) to distinguish examples belonging to shared classes. These approaches are further incorporated into domain-matching frameworks like adversarial training (You et al., 2019; Fu et al., 2020; Lifshitz & Wolf, 2021) and self-training (Chen et al., 2022a; Chang et al., 2022; Lu et al., 2024).

While PDM methods for UniDA appear promising, they have not been *comprehensively* tested on a broad spectrum of UniDA scenarios. In particular, most UniDA works follow the experimental protocols established by You et al. (2019) and Fu et al. (2020), as shown in Figure 1. The protocols examined the region of relatively higher portions of target-private classes and lower portions of source-private classes. Nevertheless, they did not test UniDA methods on high portion of source-private classes or some of the boundary cases of partial DA (X-axis) and open-set DA (Y-axis). The cases of higher source-private portions, named *extreme UniDA*, are especially challenging yet under-explored by the community. Such scenarios naturally arise in practice when models are pre-trained on large datasets and then adapted to more specialized tasks with fewer categories. In fact, our careful investigation in Figure 2 surprisingly reveals that all SOTA PDM methods for UniDA failed in extreme UniDA, falling behind the simplest baseline of training on source data only (SO). That is, existing UniDA solutions have yet to *fully* address the challenges of achieving robust performance across arbitrary divisions of label sets.

In this work, we attempt to fully achieve comprehensive UniDA by delving into the question: *What caused the performance degrade of PDM in extreme UniDA?* We observe that the abundance of source-private classes in extreme UniDA, combined with the structural disparity between source and target data when the model is trained exclusively on source-labeled data, causes target representations to experience dimensional collapse (DC) (Jing et al., 2022). This collapse degrades the quality of target representations, which in turn weakens the importance weight functions essential

for PDM's effectiveness. Consequently, negative transfer intensifies, resulting in performance that falls below SO.

To address the issue of target DC, we propose incorporating unlabeled target data into training alongside source-labeled data. Instead of employing pseudo-labeling techniques on unlabeled target data, which are inherently susceptible to the DC issue like PDM, we opt to explore the potential of *self-supervised learning* (SSL) as an alternative. Several traditional SSL techniques based on pretext tasks, such as jigsaw puzzles and rotation, have been explored for related UDA problems (Sun et al., 2019; Xu et al., 2019; Bucci et al., 2019; 2021). However, these methods are not designed to tackle the DC issue, resulting in poorly expressive representations (Wallace & Hariharan, 2020).

In contrast, we explore *different* modern SSL techniques that capitalize on two intuitions. First, semantically similar examples should lie closer in the representation space. Second, examples in the representation space should be spread out, which avoids degenerate solutions of mapping every example to the same representation and tackles DC directly. We adopt the framework of Wang & Isola (2020), demonstrating that the former can be realized by an alignment loss, and the latter can be achieved with a uniformity loss.

Our rigorous ablation study on the two SSL losses reveals a novel insight. When no SSL loss is applied to the target data (SO) or when only the alignment loss is used, the DC issue persists, resulting in the worst performance. In contrast, introducing the uniformity loss yields significant improvements in both dimensional richness and UniDA performance. Notably, uniformity and alignment losses offer distinct benefits to the UniDA process, and their combination introduces a powerful enhancement that systematically strengthens state-of-the-art PDM methods. Through an extensive range of experiments, we validate the robustness of this enhancement across diverse prior distributions of label sets. Our findings establish a new milestone and benchmark for advancing UniDA across all scenarios comprehensively.

Our contributions can be summarized as follows:

1. We are the first to highlight the unnoticed problem of extreme UniDA that prevents the community from solving UniDA comprehensively, promoting a new research direction for the community.

2. We identify the DC issue behind state-of-the-art PDM methods with careful ablation study and analysis, offering novel and fundamental understanding on the (extreme) UniDA problem.

3. We effectively resolve the DC issue by integrating underexplored SSL techniques into UniDA, establishing new SOTA performance on more comprehensive benchmarks that cover all UniDA scenarios.

## 2. Preliminaries

Universal Domain Adaptation (UniDA) (You et al., 2019) comes with a labeled source dataset $\mathcal{D}_s = \{(\mathbf{x}_i^s, y_i^s)\}_{i=1}^{n_s}$ and an unlabeled target dataset $\mathcal{D}_t = \{\mathbf{x}_i^t\}_{i=1}^{n_t}$. Each $\mathbf{x}$ represents an input vector, and $y$ denotes a discrete label. The datasets $\mathcal{D}_s$ and $\mathcal{D}_t$ are sampled from some unknown source and target distributions $p_s$ and $p_t$, respectively. We denote the label sets in the source and target domains by $\mathcal{C}_s$ and $\mathcal{C}_t$. Their intersection $\mathcal{C} = \mathcal{C}_s \cap \mathcal{C}_t$ representing the shared label set. The source-private and target-private label sets are defined as $\overline{\mathcal{C}}_s = \mathcal{C}_s \setminus \mathcal{C}$ and $\overline{\mathcal{C}}_t = \mathcal{C}_t \setminus \mathcal{C}$, respectively. The task of UniDA is to learn a feature extractor $\theta_f$ and a label classifier $\theta_c$ that classify target data to $|\mathcal{C}| + 1$ classes, where target-private classes are regarded as one unseen class. UniDA is particularly challenging as it aims to simultaneously achieve high accuracy on shared and unseen target private classes *irrespective of the prior distribution of label sets*, which can be characterized by the relative proportions of the source-private, shared and target-private classes: $\{\frac{|\overline{\mathcal{C}}_s|}{|\mathcal{C}_s \cup \mathcal{C}_t|}, \frac{|\mathcal{C}|}{|\mathcal{C}_s \cup \mathcal{C}_t|}, \frac{|\overline{\mathcal{C}}_t|}{|\mathcal{C}_s \cup \mathcal{C}_t|}\}$. Within the three ratios that naturally sum to 1, the source-private ratio $\pi_s = \frac{|\overline{\mathcal{C}}_s|}{|\mathcal{C}_s \cup \mathcal{C}_t|}$ and target-private ratio $\pi_t = \frac{|\overline{\mathcal{C}}_t|}{|\mathcal{C}_s \cup \mathcal{C}_t|}$ are often taken as the key characteristics of the UniDA scenario, as Figure 1 showed.

### 2.1. Partial Domain Matching (PDM)

In UniDA, the simplest baseline is to train the feature extractor $\theta_f$ and label classifier $\theta_c$ on source labeled data only (SO) with cross-entropy loss:

$$\mathcal{L}_s(\theta_f, \theta_c) = \mathbb{E}_{(\mathbf{x}, y) \sim p_s} \mathrm{CE}(y, \theta_c(\theta_f(\mathbf{x}))) \qquad (1)$$

To mitigate the distribution shift between source and target domains, domain matching techniques align feature distributions across domains. Adversarial training (Ganin et al., 2016) is a prominent approach in the UDA literature and is extensively employed in UniDA (You et al., 2019; Fu et al., 2020; Lifshitz & Wolf, 2021). Another widely used method in UniDA is self-training (Zou et al., 2018), which typically involves assigning pseudo-labels (Chen et al., 2022a; Lu et al., 2024; Chang et al., 2022) or leveraging entropy minimization (Saito et al., 2020).

To align feature distributions in the adversarial domain matching framework, a domain discriminator $\theta_d$ is introduced, and the adversarial loss is formulated as:

$$\mathcal{L}_{\mathrm{adv}}(\theta_f, \theta_d) = -\mathbb{E}_{\mathbf{x} \sim p_s} w_s(\mathbf{x}) \log \theta_d(\theta_f(\mathbf{x})) \qquad (2)$$
$$- \mathbb{E}_{\mathbf{x} \sim p_t} w_t(\mathbf{x}) \log(1 - \theta_d(\theta_f(\mathbf{x}))),$$

where $0 \leq w_s(\mathbf{x}), w_t(\mathbf{x}) \leq 1$ are used to downweight private-class samples. The overall objective involves minimizing cross-entropy loss on source data and solving a min-max optimization problem for domain matching:

$$\min_{\theta_f, \theta_c} \max_{\theta_d} \left[ \mathcal{L}_s(\theta_f, \theta_c) - \lambda_{\mathrm{adv}} \mathcal{L}_{\mathrm{adv}}(\theta_f, \theta_d) \right], \qquad (3)$$

where $\lambda_{\mathrm{adv}}$ is the weighted hyperparameter.

Alternatively, self-training applies cross-entropy loss to high-confidence target samples with pseudo labels:

$$\mathcal{L}_{\mathrm{ST}} = \mathbb{E}_{x \sim p_t} \mathbb{I}\{w_t(\mathbf{x}) \geq \delta\} \cdot \mathrm{CE}(y, \theta_c(\theta_f(\mathbf{x}))), \qquad (4)$$

where $\delta$ is a confidence threshold. The overall objective can be formulated as:

$$\min_{\theta_f, \theta_c} \mathcal{L}_s(\theta_f, \theta_c) + \lambda_{\mathrm{ST}} \mathcal{L}_{\mathrm{ST}}(\theta_f, \theta_c). \qquad (5)$$

where $\lambda_{\mathrm{ST}}$ is the weighted hyperparameter.

**Design of importance weight functions.** To achieve robust performance in UniDA, it is crucial to design effective importance weight functions, $w_s(\mathbf{x})$ and $w_t(\mathbf{x})$ to distinguish shared-class and private-class samples. These functions downweight private samples during domain matching by assigning a weight of 0 to private-class samples and 1 to shared-class samples.

Prior work has explored various approaches to estimate importance weights. One common approach uses uncertainty scores derived from the classifier's output (*logits*) (You et al., 2019; Fu et al., 2020; Lifshitz & Wolf, 2021; Chang et al., 2022), leveraging the notion that models trained on source data will exhibit higher uncertainty when presented with target-private data. However, since classifiers may overfit to classification tasks, an alternative line of research leverages distance metrics derived from the feature extractor's output (*representations*) (Saito et al., 2020; Li et al., 2021; Chen et al., 2022a; Lu et al., 2024), based on the assumption that shared-class examples lie near each other in the embedding space. While these importance weight functions do offer performance gains, we show that they have an inherent limitation, which we discuss in detail in the following section.

## 3. Limitations of PDM

### 3.1. Pitfalls of PDM in UniDA

The goal of UniDA is to achieve robust performance across arbitrary division of label sets. However, we observe that existing works mainly evaluate a limited range of distributions, neglecting regions with high $\pi_s$, as shown in Figure 1. Surprisingly, our comprehensive experiments (Figure 2) demonstrate that PDM methods fail to outperform SO in high $\pi_s$ region.

The inferior performance of PDM approaches suggests that they fail to benefit from the additional domain matching loss

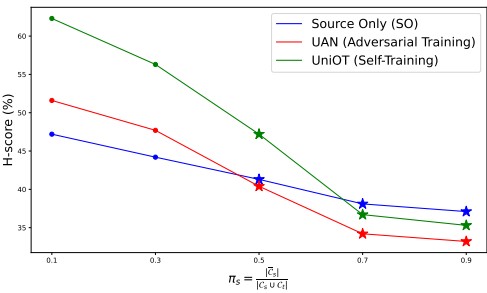

*Figure 2.* **Evaluation of Various PDM Paradigms.** We report results on the OfficeHome (Venkateswara et al., 2017) across a range of $\pi_s$ values. The star indicates previously unexplored settings.

when $\pi_s$ is high. As discussed in Section 2.1, the success of PDM relies on accurately estimating importance weights, which are derived from either logits or representations.

We hypothesize that, under a high $\pi_s$ regime, training exclusively with $\mathcal{L}_s$ may fail to capture target representations effectively due to substantial discrepancies between the source and target domains. The resulting degraded representations compromise the accuracy of importance weight functions, ultimately impairing overall performance.

To validate our hypothesis, we examine the target representations from the model trained with $\mathcal{L}_s$ under varying $\pi_s$ in Section 3.2. Next, in Section 3.3, we analyze how the degraded representation affects the performance of PDM.

### 3.2. Dimensional Collapse in Extreme UniDA

Dimensional Collapse (DC) (Jing et al., 2022) refers to the reduction in the effective dimensionality of learned representations, where features collapse onto a lower-dimensional subspace, leading to a loss of diversity in the representation space. It is usually analyzed with the singular value spectrum (Jing et al., 2022; Shi et al., 2022; Zhang et al., 2023). Further details are provided in Appendix A.

To verify that the model trained with $\mathcal{L}_s$ can significantly affect representation quality in scenarios with high $\pi_s$, we conduct singular value spectrum analysis under different values of $\pi_s$. As illustrated in Figure 3(a), when $\pi_s$ increases to 0.6, several singular values approach zero, showing a classic case of dimensional collapse (DC). This effect becomes more pronounced at $\pi_s = 0.75$, indicating that applying source-supervised loss in high $\pi_s$ scenarios exacerbates DC.

To further validate that the root cause lies in the dominance of source data, leading the model trained with $\mathcal{L}_s$ to fail in capturing the intrinsic structure of the target representation, we design an experiment using cross-entropy loss on the target data ($\mathcal{L}_t$, assuming target labels are available). Specifically, we gradually reduce the number of labeled instances in the target dataset and analyze the impact of this

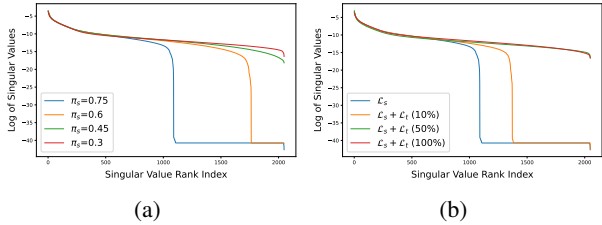

*Figure 3.* **Singular Value Spectrum Analysis**: (a) Spectrum plots for the target representations of SO ($\mathcal{L}_s$) under different choices of $\pi_s$. (b) Spectrum plots for the target representations derived from $\mathcal{L}_s$ combined with $\mathcal{L}_t$, across varying amounts of labeled target data under $\pi_s = 0.75$. DC is evidenced by certain singular values converging toward zero.

process on the quality of the target representations. From Figure 3(b), we observe that training with $\mathcal{L}_t$ on 10% of the target data significantly mitigates DC, while using 50% of the data achieves performance nearly equivalent to utilizing the entire target dataset. These results suggest that relying solely on $\mathcal{L}_s$ at high $\pi_s$ fails to capture the intrinsic structure of the target data, leading to DC.

### 3.3. Degraded Representation Quality Impairs PDM

In Section 3.2, we demonstrated that the quality of target representations deteriorates as $\pi_s$ increases. Next, we investigate how this degradation affects PDM. As discussed in Section 2.1, the importance weight function is designed to reduce the influence of private samples during domain matching. Since the weight function is computed based on representations or logits, we hypothesize that the degraded representation quality adversely impacts the effectiveness of the importance weight function.

To investigate this, we design a metric to evaluate the error rate of the importance weight function in classifying samples as shared or private. For a batch $B$ during training, the error rate is computed as:

$$E_{\text{IW}}(B) = \frac{1}{|B|} \sum_{(\mathbf{x},y) \in B} \mathbb{I}\{\hat{y}(\mathbf{x}) \neq \mathbb{I}\{y \in \mathcal{C}\}\}, \quad (6)$$

where $\hat{y}(\mathbf{x}) = \mathbb{I}\{w(\mathbf{x}) \geq 0.5\}$, $w(\mathbf{x}) \in \{w_s(\mathbf{x}), w_t(\mathbf{x})\}$ and $0 \leq w_s(\mathbf{x}), w_t(\mathbf{x}) \leq 1$.

We first examine the error-rate threshold beyond which PDM provides no improvement over SO. To do so, we run a controlled experiment where we assume perfect knowledge of shared and private classes and systematically vary the error rate in importance weights. When $\pi_s = 0.75$ (Figure 4(a)), PDM only outperforms SO if the error rate is below 0.2. Conversely, when $\pi_s = 0.25$ (Figure 4(c)), PDM can tolerate a higher error rate, remaining beneficial up to around 0.35.

Next, we investigate how importance weights, estimated ei-

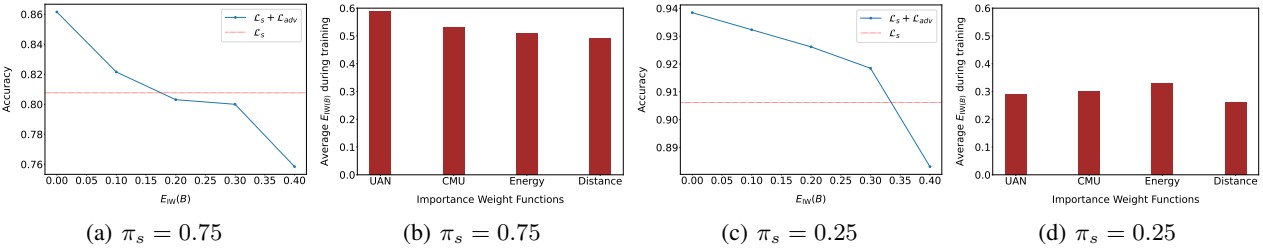

Figure 4. **Importance Weight Functions Analysis** on OfficeHome: (a)(c) show the shared-class accuracy under different $E_{\text{IW}}(B)$ compared to SO. (b)(d) present the average $E_{\text{IW}}(B)$ under common scoring functions such as uncertainty and distance.

ther from uncertainty scores or relative distances, can introduce errors in real-world scenarios where the true label set remains unknown. Specifically, we adopt two uncertainty-based methods, UAN (You et al., 2019) and CMU (Fu et al., 2020), which represent, respectively, the first application of uncertainty scores and a subsequent approach that aggregates multiple such measures. We also include the energy score (Liu et al., 2020), a highly effective out-of-distribution detection method that we found to yield the best uncertainty estimates. For distance-based approaches, we employ the L2-distance with a memory bank to compute the importance weights (Saito et al., 2020; Chen et al., 2022a).

At $\pi_s = 0.75$ (Figure 4(b)), the error rate of estimated weights surpasses 0.5—well above the 0.2 threshold needed for PDM to excel. In contrast, for $\pi_s = 0.25$ (Figure 4(d)), the error rate is closer to 0.3, which remains below the 0.35 threshold that PDM can handle effectively. The results indicate that when the representation quality substantially degrades, the specific design of importance weight functions plays only a minor role.

## 4. Tackling Dimensional Collapse with Self-Supervised Learning

In this section, we present our approach to mitigating dimensional collapse in target representations. As discussed in Section 3.2, incorporating target data can preserve structural information and alleviates DC. To achieve similar results in the absence of labels, we propose leveraging self-supervised learning (SSL). Although SSL has been widely adopted in domain adaptation tasks, including closed-set (Sun et al., 2019; Xu et al., 2019) and partial domain adaptation (Bucci et al., 2019; 2021), existing approaches predominantly rely on pretext tasks, such as jigsaw puzzles and rotation prediction (Table 1). These methods, however, are neither designed to address DC nor effective at tackling it (Wallace & Hariharan, 2020). Moreover, they have shown limited ability to generalize to downstream tasks (Teng et al., 2022; Albelwi, 2022).

In Section 4.1, we explore the usage of more advanced SSL methods and analyze the role of SSL components in UniDA.

Next, in Section 4.2, we demonstrate how the improved representations benefit PDM. Finally, we present a miniature experiment in Section 4.3 as a visualization to illustrate the effects of the proposed loss functions, providing a clearer understanding of their impact.

### 4.1. The Role of SSL Components in UniDA

In SSL, early work on pretext tasks was driven by the intuition that auxiliary tasks can enrich representations by providing diverse learning signals. However, pretext tasks are often limited by task-specific biases, which can hinder the generalization of learned representations (Wallace & Hariharan, 2020). We study the more recent methods build on the principle that similar samples should share similar representations. A straightforward approach to achieve this is to align the representations of a pair of samples generated from the same input $\mathbf{x}$ through independent augmentations. The alignment loss can be expressed as:

$$\mathcal{L}_{\text{Align}}(\theta_f) = \mathbb{E}_{\mathbf{x} \sim p} ||\theta_f(\mathcal{T}(\mathbf{x})) - \theta_f(\mathcal{T}'(\mathbf{x}))||_2^2, \quad (7)$$

where $\mathcal{T}$ and $\mathcal{T}'$ are independent random augmentation functions. However, optimizing only the alignment loss could also lead to DC (Jing et al., 2022), where the model converges to a trivial solution by collapsing all representations into a single point. To avoid this problem, prior work employs various auxiliary objectives, such as contrastive learning (Wang & Isola, 2020; Chen et al., 2020), asymmetric models (Grill et al., 2020; Chen & He, 2021) and redundancy reduction (Bardes et al., 2021; Zbontar et al., 2021).

To delve into how SSL can be applied to resolve DC, we analyze the approach proposed by Wang & Isola (2020). Their method is particularly suitable for analysis because its design can be decoupled from alignment loss, unlike asymmetric models or redundancy reduction techniques, which cannot be easily isolated. Wang & Isola (2020) introduce the concept of uniformity loss (Wang & Isola, 2020; Wang & Liu, 2021; Fang et al., 2024), which encourages the learned representations to be uniformly distributed on the unit hypersphere. It is defined using the average pairwise

*Table 1.* **Comparison of existing SSL approaches in UDA**. S and T refer to the source and target domains, respectively, while Cont. stands for contrastive learning.

| | Setting | Method | Applied Data | Goal |
|---|---|---|---|---|
| Sun et al. (2019); Xu et al. (2019) | Closed DA | Pretext | S+T | Minimize domain gap |
| Bucci et al. (2019; 2021) | Partial DA | Pretext | T | Learn target info |
| Ours | UniDA | Cont. & Non-Cont. | T | Tackle DC |

*Table 2.* Performance comparison on Office31 (A2W) with different combinations of loss functions.

| Method | H-score (%, ↑) |
|---|---|
| $\mathcal{L}_s$ | 64.5 |
| $\mathcal{L}_s + \mathcal{L}_{\text{Align}}$ | 64.6 |
| $\mathcal{L}_s + \mathcal{L}_{\text{Uniform}}$ | 65.1 |
| $\mathcal{L}_s + \mathcal{L}_{\text{Align}} + \mathcal{L}_{\text{Uniform}}$ | 67.2 |

Gaussian potential (Cohn & Kumar, 2007) as follows:

$$\mathcal{L}_{\text{Uniform}}(\theta_f) = \log \mathbb{E}_{\mathbf{x},\mathbf{x}' \overset{i.i.d}{\sim} p} [e^{-t||\theta_f(\mathbf{x})-\theta_f(\mathbf{x}')||_2^2}], \quad (8)$$

where $t$ is a fixed hyperparameter.

We study the effect of applying alignment loss and uniformity loss in the context of extreme UniDA. As illustrated in Figure 5 and Table 2, uniformity loss alone significantly alleviates DC and yields improvements, whereas alignment loss alone fails to address DC and provides little to no benefit. Combining both losses achieves the best performance. These findings suggest that preserving the intrinsic structure of the target data through uniformity loss plays a critical role in extreme UniDA. Additionally, maintaining feature invariance via alignment loss further enhances the intra-class relationships, leading to improved overall performance.

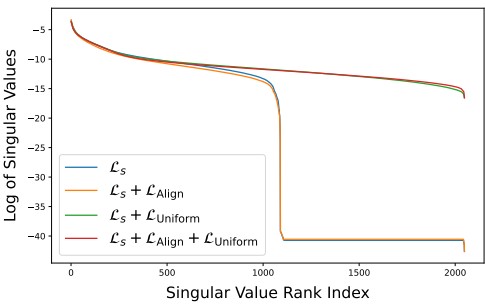

*Figure 5.* **Singular Value Spectrum Analysis**: Spectrum plot of the target representation under $\pi_s = 0.75$ with different loss function combinations. Results show that uniformity loss alone can tackle DC.

Beyond the presented framework, we also demonstrate that asymmetric models and redundancy reduction can achieve performance comparable to Wang & Isola (2020). The corresponding results are provided in Appendix B.3.

### 4.2. Improved Representation Quality Enhances PDM

In this section, we demonstrate that enhancing representation quality by SSL can further improve PDM. Figure 6 illustrates the error rate ($E_{\text{IW}}(B)$) of importance weighting in adversarial loss during training. Without incorporating the self-supervised loss, the error rate increases, resulting in performance inferior to SO. In contrast, incorporating the self-supervised loss not only improves representation quality but also reduces the error rate. These findings suggest that enhancing representation quality is more critical for improving PDM than refining the design of the importance weight functions.

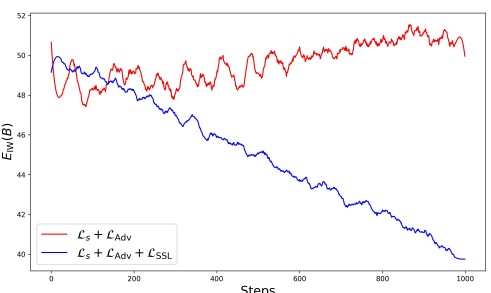

*Figure 6.* **Analysis of Error Rate in Importance Weight Function**: The error rate decreases when SSL is applied.

### 4.3. Miniature Experiment Analysis

To further study the effect of different loss functions under varying $\pi_s$, we conduct a miniature experiment to visualize the impact. Inspired by Liu et al. (2022), we generate a 2D dataset simulating scenarios with different values of $\pi_s$, as illustrated in Figure 7(a) and 7(c). This experiment aims to illustrate two key points: (1) the DC problem encountered when training with $\mathcal{L}_s$ in extreme UniDA, as discussed in Section 3.2, and (2) the effect of SSL in tackling DC in extreme UniDA as discussed in Section 4.1.

The dataset consists of three classes: the blue and green classes represent shared classes, distinguished by shape to indicate their respective domains. The red class represents source-private data, and its distribution shape simulates a mixture of source-private classes, with the numbers indicating their respective class counts. Further details about the settings are provided in Appendix D.5.

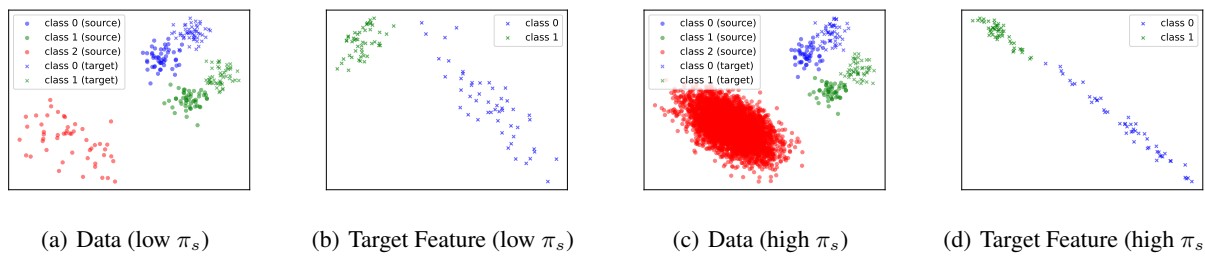

(a) Data (low $\pi_s$)      (b) Target Feature (low $\pi_s$)      (c) Data (high $\pi_s$)      (d) Target Feature (high $\pi_s$)

*Figure 7.* **Toy experiment visualization with `SO` under different values of** $\pi_s$: Circles and crosses represent source and target data, respectively. The number of red points simulates the "extremity" in universal domain adaptation. (a)(c) show the original data. (b)(d) show the target features.

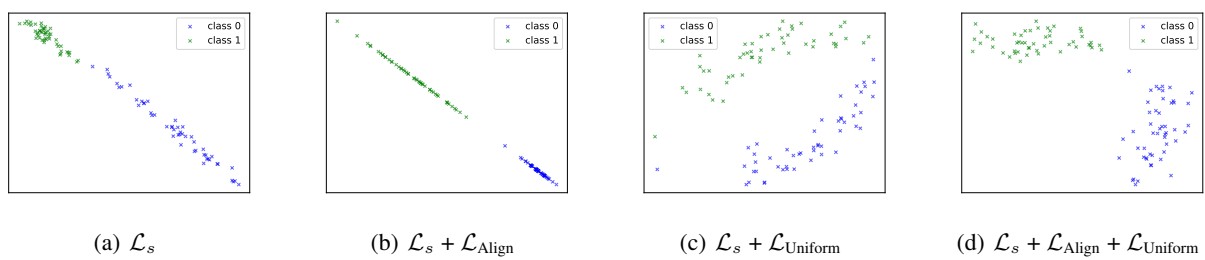

(a) $\mathcal{L}_s$      (b) $\mathcal{L}_s + \mathcal{L}_{\text{Align}}$      (c) $\mathcal{L}_s + \mathcal{L}_{\text{Uniform}}$      (d) $\mathcal{L}_s + \mathcal{L}_{\text{Align}} + \mathcal{L}_{\text{Uniform}}$

*Figure 8.* **Toy experiment visualization with different SSL components under different values of** $\pi_s$: All images show target features.

Figures 7(b) and 7(d) present the results of training with $\mathcal{L}_s$. The representations exhibit a collapse into a line under high $\pi_s$, whereas they retain their structure under low $\pi_s$. To further study the effect of different components of SSL in high $\pi_s$ scenarios, we visualize the results in Figure 8. The $\mathcal{L}_{\text{Align}}$ term encourages tighter clustering of representations but exacerbates collapse. On the other hand, $\mathcal{L}_{\text{Uniform}}$ mitigates collapse at the expense of weaker within-class aggregation. Combining both terms offers a balanced solution, preserving structural distinctions while still promoting meaningful within-class clustering.

## 5. Experiments

Our experiments seek to answer the following questions: (1) Can SSL benefit all PDM methods? (2) Can SSL generalize across all scenarios, regardless of the prior distribution of label sets?

### 5.1. Experimental Setup

**Dataset.** We present results on four widely used benchmarks: Office31, OfficeHome, VisDA, and DomainNet. Details of these datasets can be found in Appendix D.3.

**Evaluation metric.** We adopt H-score (Fu et al., 2020) as the metric, which calculate the harmonic mean of accuracy on common classes $a_{\mathcal{C}}$ and accuracy on target-private (unseen) classes $a_{\overline{\mathcal{C}}_t}$. See Appendix D.2 for details.

**Baselines.** We considered methods from various domain-matching frameworks and different importance weight functions. Specifically, `UAN` and `CMU` follow adversarial learning paradigms, while `UniOT`, `DANCE`, and `MLNet` adopt self-training strategies. Further details on these methods are provided in Appendix C.1.

**Extreme UniDA setting.** Considering the challenges of evaluating all methods across every possible label set division on all datasets, we revealed a setting with a high $\pi_s$ to specifically evaluate the effectiveness of PDM methods. In this setup, $\pi_s$ is set to values greater than 0.65 across all datasets, in contrast to the original settings where $\pi_s$ is below 0.35. Further details about this setting can be found in Appendix D.1.

### 5.2. Discussion

**Compatibility with different PDM methods.** In Tables 3 and 4 (Appendix B.1), we present results for two training paradigms under the extreme UniDA setting ($\pi_s > 0.65$). Our findings indicate that the proposed approach improves performance in both paradigms, with a more pronounced effect under adversarial training. We attribute this discrepancy to adversarial training's explicit effort to match the distribution of selected features, making it more sensitive to errors in the importance weight function. In contrast, self-training mainly assigns pseudo-labels to high-confidence samples, resulting in a less pronounced improvement.

*Table 3.* H-score (%, ↑) on **Office** and **DomainNet**. For each column, the best values are highlighted in **bold**, while the top value in each category is highlighted with underline. **IW** refers to the calculation of importance weight functions. **UM**: uncertainty measurement; **OT**: assignment matrix computed via optimal transport; **NN**: nearest neighbor determined by relative distance in the embedding space.

| Method | IW | Office | | | | | | | DomainNet | | | | | | |
| --- | --- | --- | --- | --- | --- | --- | --- | --- | --- | --- | --- | --- | --- | --- | --- |
| | | A2D | A2W | D2A | D2W | W2A | W2D | Avg | P2R | R2P | P2S | S2P | R2S | S2R | Avg |
| **Adversarial Training** | | | | | | | | | | | | | | | |
| UAN (You et al., 2019) | UM | 24.5 | 61.8 | 48.9 | 64.2 | 27.9 | 61.3 | 48.1 | 11.9 | 15.1 | 14.4 | 17.2 | 18.1 | 11.3 | 14.6 |
| CMU (Fu et al., 2020) | UM | 76.8 | 63.8 | 56.1 | 77.2 | 66.3 | 78.2 | 69.7 | 30.1 | 42.4 | 34.1 | 24.3 | 32.2 | 34.1 | 32.8 |
| **UAN+ SSL** | UM | **87.4** | 74.9 | 72.4 | **81.3** | 74.9 | **87.7** | 79.8 | **50.1** | 39.2 | 35.9 | 32.7 | 34.0 | 49.8 | 40.3 |
| **Self-Training** | | | | | | | | | | | | | | | |
| UniOT (Chang et al., 2022) | OT | 78.8 | 67.7 | **86.1** | 66.9 | 83.8 | 81.0 | 77.4 | 38.1 | 29.8 | 30.8 | 29.3 | 29.1 | 38.3 | 32.6 |
| **UniOT+ SSL** | OT | 79.8 | **75.9** | 86.0 | 77.3 | **84.1** | 82.4 | **80.9** | 39.6 | 29.9 | 33.6 | 31.4 | 31.1 | 40.2 | 34.3 |
| DANCE (Saito et al., 2020) | NN | 49.7 | 47.9 | 48.4 | 54.9 | 48.9 | 55.6 | 50.9 | 39.4 | 3.30 | 11.8 | 0.90 | 7.60 | 35.3 | 16.4 |
| MLNet (Lu et al., 2024) | NN | 51.2 | 61.9 | 58.1 | 79.2 | 59.5 | 75.5 | 64.2 | 48.0 | 45.9 | 47.8 | 48.5 | 43.7 | **53.0** | 47.8 |
| **MLNet+ SSL** | NN | 48.6 | 60.5 | 60.2 | 80.6 | 60.7 | 80.3 | 65.2 | 48.8 | **46.5** | **50.2** | **50.6** | **44.7** | 52.5 | **48.9** |

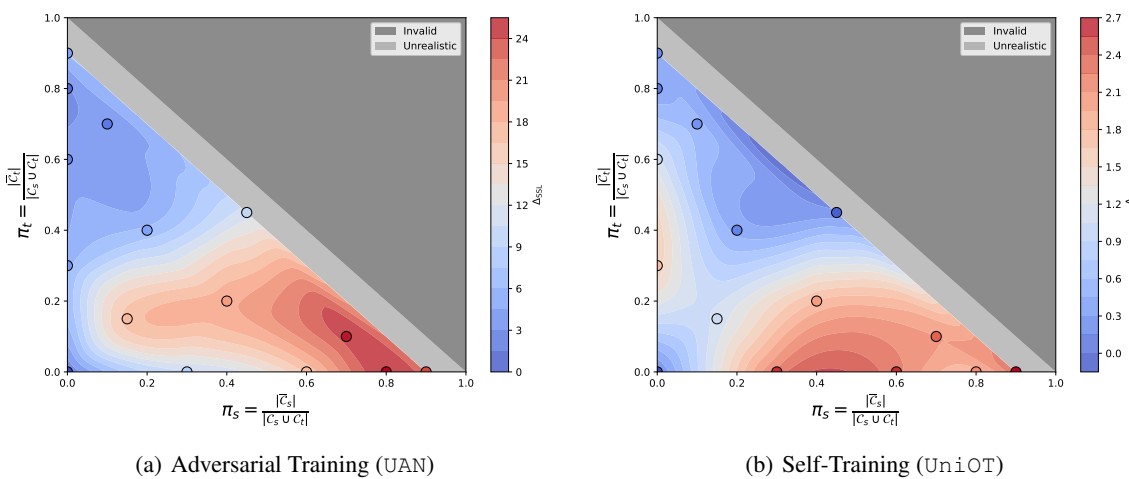

(a) Adversarial Training (UAN)        (b) Self-Training (UniOT)

*Figure 9.* **Improvement of SSL on the UniDA Spectrum**: It shows the effectiveness of SSL across different $(\pi_s, \pi_t)$ values within the UniDA spectrum on two different training paradigms.

For completeness, we also evaluated our approach in general UniDA setting ($\pi_s < 0.35$) in Tables 5 and 6 (both in Appendix B.1). The results suggest that it continues to provide a modest improvement. These findings highlight the compatibility of SSL with various PDM methods across both ends of the UniDA spectrum ($\pi_s > 0.65$ and $\pi_s < 0.35$). Moreover, the performance gap between extreme and general settings indicates that SSL primarily addresses the DC issue, as discussed in the next paragraph.

**Robustness regardless of prior label set distributions** In this section, we evaluate the effect of SSL across varying values in the UniDA spectrum. Figure 9 illustrates the results for eight different combinations of $(\pi_s, \pi_t)$, with the corresponding contour plot showing the improvement introduced by SSL ($\Delta_{\text{SSL}}$). The results demonstrate that SSL

consistently improves performance across all divisions of label sets, indicating its robustness. Moreover, the improvement becomes more pronounced as $\pi_s$ increases (toward the lower-right region), which corroborate our analysis that SSL is particularly effective at addressing DC, a challenge that is especially severe in high $\pi_s$ scenarios.

## 6. Conclusion

In this work, we underline extreme UniDA, a challenging and under-explored scenario that blocks state-of-the-art PDM methods toward solving UniDA comprehensively. We identify dimensional collapse in target representations as the primary cause of PDM's poor performance, arising when the private source data overwhelms the training process. This collapse subsequently undermines the effectiveness of

partial domain matching. Motivated by these findings, we propose addressing dimensional collapse from the perspective of target data. To achieve this, we revisit self-supervised learning and uncover the critical role of uniformity in mitigating dimensional collapse. Our proposed solution is compatible with various partial domain matching methods and accommodates a broad range of label set distributions.

## Acknowledgments

We thank the anonymous reviewers and members of CLLab for their constructive feedback. This work is supported by the National Science and Technology Council in Taiwan via NSTC 113-2628-E-002-003 and NSTC 113-2634-F-002-008. We thank to National Center for High-performance Computing (NCHC) of National Applied Research Laboratories (NARLabs) in Taiwan for providing computational and storage resources.

## Impact Statement

This paper provides a more comprehensive investigation of domain adaptation, enabling models to generalize across a broader range of scenarios. This research does not pose significant societal consequences.

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

# A. Additional Related Work

Most related work is discussed in each section, and a more detailed version is provided below.

## A.1. Self-Supervised Learning for Domain Adaptation

Self-supervised learning (SSL) has been widely adopted in various unsupervised domain adaptation (UDA) tasks due to its ability to capture invariant features that help bridge domain gaps. Xu et al. (2019) first demonstrated its effectiveness in unsupervised domain adaptation through simple pretext tasks for object detection and semantic segmentation. Building on this foundation, Bucci et al. (2019; 2021) extended pretext tasks to partial domain adaptation. However, these methods are not designed to address DC and have been shown to exhibit limited expressiveness due to task-specific biases (Wallace & Hariharan, 2020). Our work extends the application of SSL to resolving DC in extreme UniDA by exploring more recent methods, including contrastive (Wang & Isola, 2020; Chen et al., 2020) and non-contrastive approaches (Grill et al., 2020; Chen & He, 2021; Bardes et al., 2021; Zbontar et al., 2021). DANCE (Saito et al., 2020) addresses UniDA through clustering based on source data. Unlike DANCE, our approach leverages SSL on target data independently of source data, resulting in significantly improved performance, as shown in Section 5. SSL has also been explored for UDA in point cloud tasks, where Achituve et al. (2021) applied deformation reconstruction to align representations across domains.

Beyond UDA, recent studies highlight the benefits of SSL pretraining in settings involving distribution shifts or imbalanced learning. Garg et al. (2024) showed that combining self-training with contrastive SSL pretraining outperforms either approach alone. Similarly, Liu et al. (2022) found that SSL enhances representation learning for minority classes, improving robustness in imbalanced learning scenarios.

## A.2. Universal Domain Adaptation

Universal domain adaptation (UniDA) is a more general form of UDA that makes no assumptions about the label sets relationship between the source and target domains. To achieve domain matching without the interference of private-class data. Prior works have leveraged uncertainty measurement (You et al., 2019; Fu et al., 2020; Lifshitz & Wolf, 2021; Chen et al., 2022b), assignment matrices obtained from optimal transport (Chang et al., 2022), and nearest-neighbor methods based on relative distance (Saito et al., 2020; Lu et al., 2024) as importance weight functions to distinguish shared and private classes for domain matching. While these methods focus on designing various importance weight functions, our work takes a different approach. We demonstrate that the unsatisfactory performance of partial domain matching stems from degraded representation quality caused by dimensional collapse, rather than the choice of importance weighting alone.

Another line of research (Saito & Saenko, 2021; Hur et al., 2023; Lu et al., 2024) focuses on developing robust open-set classifiers to distinguish between common and private classes in target data. While these methods do not explicitly address domain matching, most of them can be seamlessly integrated into any domain matching framework. With the emergence of more advanced models, Zhu et al. (2023b); Deng & Jia (2023) explore the application of models such as vision transformers and pretrained vision models like DINO (Caron et al., 2021) and CLIP (Radford et al., 2021) to UniDA. There are also works that align with our goal of exploring more realistic or under-explored scenarios in UniDA. Qu et al. (2024) investigates source-free UniDA, where source data is unavailable during adaptation. Zhu et al. (2023a) addresses generalized UniDA, which aims to identify novel categories and label distributions in the target domain, utilizing active learning to achieve this objective.

## A.3. Dimensional Collapse

Dimensional Collapse (DC) refers to the reduction in the effective dimensionality of learned representations, where features collapse onto a lower-dimensional subspace, leading to a loss of diversity in the representation space. DC has been extensively studied across various domains, including metric learning (Roth et al., 2020), self-supervised learning (Jing et al., 2022), class-incremental learning (Shi et al., 2022), federated learning with heterogeneity (Shi et al., 2023), and graph collaborative filtering (Zhang et al., 2023). Previous work primarily employs the singular value spectrum to analyze DC effects. This involves computing the covariance matrix of embeddings, applying singular value decomposition, and examining the distribution of singular values (usually in logarithmic scale) to reveal insights into the DC effects.

# B. Supplementary Experimental Results

## B.1. Compatibility with different PDM methods

Tables 3 and 4 present the results for extreme UniDA settings, while Tables 5 and 6 summarize the results for general UniDA settings. The resutls indicate SSL can be applied to various PDM methods in both general and extreme UniDA settings.

*Table 4.* H-score (%, ↑) on **Office-Home** and **VisDA**. For each column, the best values are highlighted in **bold**, while the top value in each category is highlighted with underline. **IW** refers to the calculation of importance weight functions. **UM**: uncertainty measurement; **OT**: assignment matrix computed via optimal transport; **NN**: nearest neighbor determined by relative distance in the embedding space.

| | | Office-Home | | | | | | | | | | | | VisDA |
|---|---|---|---|---|---|---|---|---|---|---|---|---|---|---|
| Method | IW | Ar2Cl | Ar2Pr | Ar2Rw | Cl2Ar | Cl2Pr | Cl2Rw | Pr2Ar | Pr2Cl | Pr2Rw | Rw2Ar | Rw2Cl | Rw2Pr | Avg | S2R |
| | | | | | | | Adversarial Training | | | | | | | |
| UAN (You et al., 2019) | UM | 29.9 | 36.4 | 14.1 | 22.4 | 20.6 | 16.4 | 26.4 | 25.1 | 27.3 | 31.3 | 24.4 | 35.4 | 25.8 | 41.5 |
| CMU (Fu et al., 2020) | UM | 38.5 | 43.5 | 45.7 | 41.4 | 41.2 | 47.5 | 46.0 | 46.6 | 40.3 | 41.5 | 38.5 | 27.2 | 41.5 | 34.1 |
| **UAN+SSL** | UM | 47.1 | **74.4** | 76.8 | 46.4 | 54.3 | 63.5 | 55.8 | 48.5 | **72.3** | 57.7 | 46.3 | **68.5** | 59.3 | **89.5** |
| | | | | | | | Self-Training | | | | | | | |
| UniOT (Chang et al., 2022) | OT | 27.2 | 32.3 | 26.6 | 28.4 | 29.9 | 23.2 | 31.4 | 29.3 | 23.0 | 35.9 | 34.3 | 35.3 | 29.7 | 49.9 |
| **UniOT+ SSL** | OT | 32.1 | 30.3 | 31.0 | 29.7 | 28.9 | 25.6 | 32.1 | 33.9 | 29.1 | 36.7 | 35.3 | 45.6 | 32.5 | 61.1 |
| DANCE (Saito et al., 2020) | NN | 34.0 | 55.5 | **82.6** | 43.4 | 44.2 | 60.1 | 34.4 | 20.8 | 61.2 | **65.7** | 33.6 | 61.7 | 44.6 | 69.1 |
| MLNet (Lu et al., 2024) | NN | 58.2 | 66.5 | 63.3 | 69.4 | 71.2 | 64.1 | 51.3 | 59.6 | 67.7 | 49.9 | **65.3** | 56.3 | 61.9 | 75.1 |
| **MLNet+ SSL** | NN | **59.4** | 71.5 | 72.9 | **71.0** | **71.5** | **66.7** | **57.5** | **60.4** | 68.8 | 59.9 | 64.4 | 53.3 | **64.8** | 80.2 |

*Table 5.* H-score(%, ↑) on **Office-Home** (5/10/50). For each column, the best values are highlighted in **bold**, while the top value in each category is highlighted with underline. **IW** refers to the calculation of importance weight functions. **UM**: uncertainty measurement; **OT**: assignment matrix computed via optimal transport; **NN**: nearest neighbor determined by relative distance in the embedding space.

| | | Office-Home (5/10/50) | | | | | | | | | | | |
|---|---|---|---|---|---|---|---|---|---|---|---|---|---|
| Method | IW | Ar2Cl | Ar2Pr | Ar2Rw | Cl2Ar | Cl2Pr | Cl2Rw | Pr2Ar | Pr2Cl | Pr2Rw | Rw2Ar | Rw2Cl | Rw2Pr | Avg |
| | | | | | | | Adversarial Training | | | | | | |
| UAN (You et al., 2019) | UM | 51.6 | 51.7 | 54.3 | 61.7 | 57.6 | 61.9 | 50.4 | 47.6 | 61.5 | 62.9 | 52.6 | 65.2 | 56.6 |
| CMU (Fu et al., 2020) | UM | 56 | 56.9 | 59.2 | 67.0 | 64.3 | 67.8 | 54.7 | 51.1 | 66.4 | 68.2 | 57.9 | 69.7 | 61.6 |
| **UAN+ SSL** | UM | 53.8 | 75.1 | 83.9 | 63.2 | 67 | 77.6 | 72.2 | 55.9 | 81.6 | 74.2 | 55.9 | 81.6 | 70.2 |
| | | | | | | | Self-Training | | | | | | |
| DANCE (Saito et al., 2020) | NN | 26.7 | 11.3 | 18.0 | 33.2 | 12.5 | 14.3 | 41.6 | 39.9 | 33.3 | 16.3 | 27.1 | 25.9 | 25.0 |
| UniOT (Chang et al., 2022) | OT | 67.3 | 80.5 | 86.0 | 73.5 | **77.3** | **84.3** | 75.5 | 63.3 | 86.0 | **77.8** | 65.4 | 81.9 | 76.6 |
| **UniOT+ SSL** | OT | **70.1** | **80.7** | **87.3** | **73.8** | 76.7 | 84.0 | **76.1** | **63.9** | **86.2** | 77.4 | **66.3** | **83.1** | **77.1** |

*Table 6.* H-score(%, ↑) on **Office** (10/10/11). For each column, the best values are highlighted in **bold**, while the top value in each category is highlighted with underline. **IW** refers to the calculation of importance weight functions. **UM**: uncertainty measurement; **OT**: assignment matrix computed via optimal transport; **NN**: nearest neighbor determined by relative distance in the embedding space.

| | | Office (10/10/10) | | | | | | |
|---|---|---|---|---|---|---|---|---|
| Method | IW | A2D | A2W | D2A | D2W | W2A | W2D | Avg |
| | | Adversarial Learning | | | | | | |
| UAN (You et al., 2019) | UM | 59.7 | 58.6 | 60.1 | 70.6 | 60.3 | 71.4 | 63.5 |
| CMU (Fu et al., 2020) | UM | 68.1 | 67.3 | 71.4 | 79.3 | 72.2 | 80.4 | 73.1 |
| **UAN+ SSL** | UM | 85.8 | 83.5 | 84.7 | 96.4 | 84.2 | **97.2** | 88.6 |
| | | Self-Training | | | | | | |
| DANCE (Saito et al., 2020) | NN | 72.6 | 62.4 | 63.3 | 76.3 | 57.4 | 82.8 | 66.6 |
| UniOT (Chang et al., 2022) | OT | **87.0** | 88.5 | 88.4 | **98.8** | 87.6 | 96.6 | 91.2 |
| **UniOT+ SSL** | OT | 86.6 | **88.8** | **90.7** | 98.2 | **88.6** | 96.7 | **91.6** |

## B.2. Results of applying SSL on different partition of target data

Since applying supervised loss to source-private data can have significant negative effects, we investigate whether applying SSL to target-private data has a similar impact. In Figure 10, the results indicate that while it does introduce some negative effects, these are relatively minor compared to the benefits it provides. This could be attributed to its role in preserving structural information rather than focusing solely on classification like supervised loss (Liu et al., 2022).

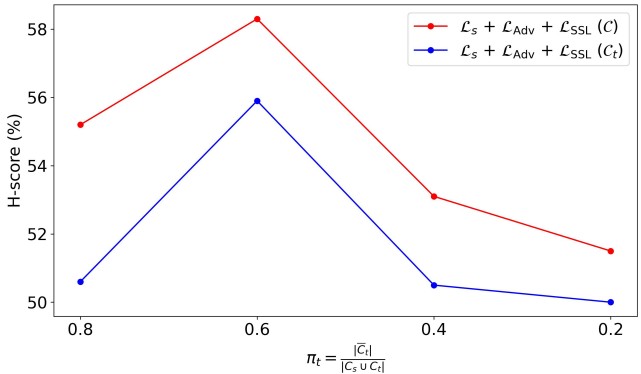

*Figure 10.* The figure illustrates the performance of applying SSL to either the shared subset of the target data or the entire target data under different $\pi_t$ values. The results suggest that applying SSL to target-private classes can indeed have a negative effect, but this impact is relatively minor compared to the overall benefits it provides

## B.3. Results of different SSL methods

We discuss three paradigms in SSL: for contrastive learning, we adopt AlignUniform (Wang & Isola, 2020); for asymmetric models, we utilize SimSiam (Chen & He, 2021); and for redundancy reduction methods, we employ Barlow Twins (Zbontar et al., 2021). As shown in Table 7, our results indicate that each of these frameworks significantly enhances performance.

*Table 7.* Performance comparison on OfficeHome (P2R) using different SSL methods.

| Method | H-score |
|---|---|
| SO | 63.2 |
| UAN | 27.3 |
| UAN + SimSiam (Chen & He, 2021) | 72.3 |
| UAN + AlignUniform (Wang & Isola, 2020) | 71.8 |
| UAN + Barlow Twins (Zbontar et al., 2021) | 70.6 |

## B.4. Results with error bar

We report the results on Office31 (Saenko et al., 2010) based on three runs in Table 8, each using a different random seed. The standard deviation values are relatively minor compared to the advantages we observe over prior works. The results indicate that our method is stable in repetitive trials.

*Table 8.* H-score of UAN + SSL with error bars on Office31.

| | A2D | A2W | D2A | D2W | W2A | W2D |
|---|---|---|---|---|---|---|
| UAN + SSL | $75.6 \pm 0.7$ | $85.4 \pm 2.6$ | $87.1 \pm 2.6$ | $76.3 \pm 1.4$ | $72.9 \pm 1.3$ | $80.0 \pm 2.5$ |

## B.5. Sensitivity of hyperparameters

We evaluated the sensitivity of the weighted hyperparameter $\lambda_{\text{SSL}}$ by experimenting values between 0.3 and 0.7. Figure 11 demonstrates minimal sensitivity to this hyperparameter across three settings in the Office-Home. The evaluations are conducted using `UAN+SSL`.

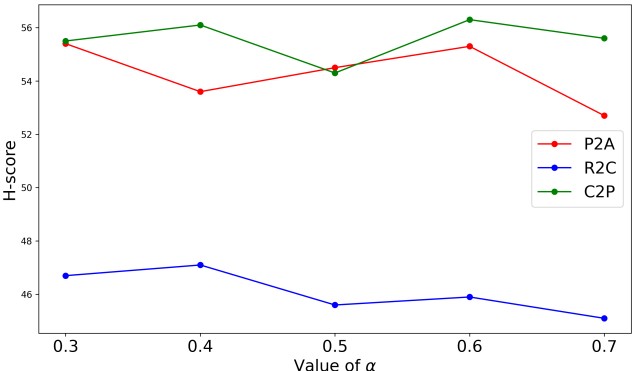

*Figure 11.* **Sensitivity of $\lambda_{\text{SSL}}$.**

# C. Supplementary Background

## C.1. Partial Domain Matching Framework

| Matching Method
Weight Calculation | Adversarial Training | Self-Training |
|---|---|---|
| **Uncertainty Measurement** | UAN, CMU, SS | - |
| **Optimal Transport** | - | UniOT |
| **Nearest Neighbor** | - | DANCE, MLNet |

*Table 9.* **PDM Framework of Different Methods.**

We introduce two main frameworks widely used in UniDA literature: adversarial training and self-training.

**Adversarial Training.** Adversarial training aligns feature distributions across domains by optimizing an adversarial loss, which encourages the feature extractor to generate domain-invariant features that fool a domain discriminator. The adversarial loss is defined as follows:

$$\mathcal{L}_{\text{adv}}(\theta_f, \theta_d) = -\mathbb{E}_{\mathbf{x} \sim p_s} w_s(\mathbf{x}) \log \theta_d(\theta_f(\mathbf{x})) - \mathbb{E}_{\mathbf{x} \sim p_t} w_t(\mathbf{x}) \log(1 - \theta_d(\theta_f(\mathbf{x}))) \tag{9}$$

Here, $w_s(\mathbf{x})$ and $w_t(\mathbf{x})$ are importance weight functions that downweight private-class samples in the source and target domains, ensuring that only shared-class samples are aligned. In an ideal case, the weight $w_s(\mathbf{x})$ and $w_t(\mathbf{x})$ should assign 0 to private-class samples and 1 to shared-class samples. The overall objective can be formulated as:

$$\min_{\theta_f, \theta_c} \max_{\theta_d} \big[ \mathcal{L}_s(\theta_f, \theta_c) - \lambda \mathcal{L}_{\text{adv}}(\theta_f, \theta_d) \big], \tag{10}$$

where $\lambda$ is the weighted hyperparameter.

**Self-Training.** Self-training approaches, unlike adversarial training, do not explicitly align the feature distributions across domains. Instead, they assign pseudo-labels to high-confidence target samples and use these pseudo-labels for training. The self-training loss is defined as:

$$\mathcal{L}_{\text{ST}}(\theta_f, \theta_c) = \mathbb{E}_{\mathbf{x} \sim p_t} \mathbb{I}\{w_t(\mathbf{x}) \geq \delta\} \cdot \mathbb{CE}(\hat{y}, \theta_c(\theta_f(\mathbf{x}))), \tag{11}$$

where $\delta$ is a confidence threshold.

The importance weight function $w_t(\mathbf{x})$ is used to filter out private-class samples and to prioritize target samples with high confidence for training. The overall objective can be formulated as:

$$\min_{\theta_f,\theta_c} \mathcal{L}_s(\theta_f, \theta_c) + \lambda_{\text{ST}}\mathcal{L}_{\text{ST}}(\theta_f, \theta_c) \tag{12}$$

### C.2. Calculation of Importance Weight Function

Let $p(y|\mathbf{x})$ represent the predicted probability distribution over the possible classes $y$ given an input $\mathbf{x}$. Specifically, $p(y_i|\mathbf{x})$ is the probability assigned to class $y_i$ for the input $x$, where $i = 1, 2, \cdots, K$ and $K$ is the number of classes. In our cases, $K = |\mathcal{C}_s|$.

**Entropy.**    The entropy $H(p)$ is defined as:

$$H(p) = -\sum_{i=1}^{K} p(y_i|\mathbf{x}) \log p(y_i|\mathbf{x}) \tag{13}$$

**Confidence.**    The confidence $C(\mathbf{x})$ is defined as the predicted probability for the most likely class:

$$C(\mathbf{x}) = \max_i p(y_i|\mathbf{x}) \tag{14}$$

**Energy Score.**    The energy score $E(\mathbf{x})$ is calculated as:

$$E(\mathbf{x}) = -\log \sum_{i=1}^{K} \exp(p(y_i|\mathbf{x})) \tag{15}$$

**Relative Distance.**    In UniDA, shared classes in the source domain are expected to be closer to shared classes in the target domain compared to target-private classes. Therefore, we can leverage this relationship to distinguish between the different label sets. In this method, clustering is first performed on the source data, and the distance from a given input $\mathbf{x}$ to the nearest cluster centroid is used to calculate the uncertainty. Let $C_j$ represent the centroid of the $j$-th cluster, and the uncertainty score $U(\mathbf{x})$ is computed as:

$$U(\mathbf{x}) = \min_j d(x, C_j), \tag{16}$$

where $d$ is a distance metric, such as Euclidean distance. The same process can be applied when the input is from the source domain. Note that the score is updated every $k$ steps, as calculate the distances in every step is costly.

## D. Details of Experimental Setup

### D.1. Extreme UniDA setting

*Table 10.* **Comparison of general and extreme settings across datasets**. The general UniDA setting refers to the conventional setup used in prior works.

| Dataset | General | | | | Extreme | | | |
|---|---|---|---|---|---|---|---|---|
| | $\|\overline{\mathcal{C}}_s\|$ | $\mathcal{C}$ | $\|\overline{\mathcal{C}}_t\|$ | $\pi_s$ | $\|\overline{\mathcal{C}}_s\|$ | $\mathcal{C}$ | $\|\overline{\mathcal{C}}_t\|$ | $\pi_s$ |
| Office-31 (Saenko et al., 2010) | 10 | 10 | 11 | 0.33 | 24 | 5 | 3 | 0.75 |
| Office-Home (Venkateswara et al., 2017) | 5 | 10 | 50 | 0.08 | 50 | 10 | 5 | 0.77 |
| Visda (Peng et al., 2017) | 3 | 6 | 3 | 0.25 | 8 | 2 | 2 | 0.67 |
| DomainNet (Peng et al., 2019) | 50 | 150 | 145 | 0.14 | 250 | 50 | 45 | 0.72 |

In Table 10, we provide the details of label set distributions for our extreme settings. Following prior work, the classes in each label set are first sorted alphabetically and then divided into three groups: source-private, common, and target-private.

## D.2. Metrics

H-score (Fu et al., 2020) is defined as the the harmonic mean of accuracy on common classes $a_{\mathcal{C}}$ and accuracy on target-private (unknown) classes $a_{\overline{\mathcal{C}}_t}$.

$$\text{H-score} = 2 \cdot \frac{a_{\mathcal{C}} \cdot a_{\overline{\mathcal{C}}_t}}{a_{\mathcal{C}} + a_{\overline{\mathcal{C}}_t}}.$$

## D.3. Dataset

Office31 (Saenko et al., 2010) contains 31 classes and three domains: Amazon (A), DSLR (D), and Webcam (W), with a total of about 4k images. Office-Home (Venkateswara et al., 2017) has 65 classes and four domains: Art (A), Product (Pr), Clipart (Cl), and Realworld (Rw), with approximately 15k images. VisDA (Peng et al., 2017) is a larger dataset with 12 classes from two domains: Synthetic and Real images, totaling around 280k images. DomainNet (Peng et al., 2019), the largest DA dataset, has 345 classes and six domains, with about 0.6 million images. Following prior works (Fu et al., 2020; Chang et al., 2022; Kundu et al., 2022), we use only three domains: Real (R), Sketch (S), and Painting (P).

## D.4. Implementation Details

We use ResNet-50 (He et al., 2016) as the backbone model for all experiments, which is pre-trained on ImageNet (Deng et al., 2009). The optimizer, scheduler and learning rate are consistent with You et al. (2019). The training steps are 10K for all experiments and the batch size is set to 36 for both domains. The hyperparameters are set as follows: $\lambda_{\text{Adv}} = 0.5$ and $\lambda_{\text{SSL}} = 0.5$ for Office-Home, DomainNet and VisDA, and $\lambda_{\text{SSL}} = 0.2$ for Office31. We use SimSiam (Chen & He, 2021) as our self-supervised loss as it does not require negative samples or large batch size. The data augmentation strategy follows the same setup as SimSiam.

## D.5. Toy Experiment Setup

The dataset comprises three classes: the blue and green classes represent shared classes, differentiated by shape to indicate their respective domains. Specifically, source classes 0, target class 0, source class 1, and target class 1 are sampled with means of $(0, 0), (3, 3), (3, -5)$ and $(6, -2)$, respectively. All shared classes have an identity covariance matrix, with 50 samples per class. The red classes represents source-private data, sampled with a mean of $(-10, -10)$ and a covariance matrix of $\begin{bmatrix} 5 & -5 \\ -5 & 1 \end{bmatrix}$. To simulate different $\pi_s$ values, the number of samples in the source-private class varies: we use 50 samples for low $\pi_s$ and 2500 samples for high $\pi_s$. For the feature extractor $\theta_f$, we use a two-layer linear network with ReLU activation (Agarap, 2018), where the output size is 2. The classifier $\theta_c$ is a single linear layer with an output size of 3, used to predict the three source classes.

