# OpenReview forum: "Tackling Dimensional Collapse toward Comprehensive Universal Domain Adaptation"
_ICML.cc/2025/Conference — ICML 2025 poster_

### Official Review · Reviewer_sBWy · 2025-03-11

**Overall Recommendation:** 3

**Summary:**

Universal Domain Adaptation (UniDA) tackles unsupervised domain adaptation where the target domain may contain classes that differ arbitrarily from the source, except for a common subset. A common approach, partial domain matching (PDM), aligns only the shared classes but often fails when many source classes are missing in the target—performing even worse than a simple baseline trained solely on source data. This work reveals that such failure is due to dimensional collapse in the target representations. To remedy this, the authors propose leveraging both alignment and uniformity techniques from modern self-supervised learning (SSL) on the unlabeled target data, preserving the intrinsic structure of the learned features. Experimental results demonstrate that SSL significantly improves PDM, setting new state-of-the-art performance across various UniDA scenarios with different proportions of shared classes, marking an important advance toward comprehensive UniDA

**Claims And Evidence:**

The claims and evidence in general are clear and convincing. But why SSL works in addressing dimension collapse may need more explanation.

**Essential References Not Discussed:**

UDA is an extension and generalization of previous methods such as PDA, OSDA, and SDA. In the Additional Related Works section, we recommend reviewing these settings as well to ensure the overall completeness of the paper.

**Experimental Designs Or Analyses:**

1. The experiments need to be further extended. UniDA is essentially a generalization of basic DA paradigms such as PDA, OSDA, and SDA, and Extreme UDA is an even broader generalization with greater challenges. If the proposed method performs well in the more challenging UDA settings, it is also necessary to evaluate it on simpler settings (like those inherent in PDA/OSDA) to demonstrate the method's robustness.

2. The ablation study is somewhat simplistic, as it was only conducted on a subtask of the small Office31 dataset, making it difficult to draw comprehensive conclusions. It is necessary to conduct experiments on a wider range of tasks to better demonstrate the method's effectiveness.

**Methods And Evaluation Criteria:**

1. There are concerns regarding the novelty of using SSL to address the dimensional collapse issue. The authors should further clarify this approach and compare it with previous SSL methods used in domain adaptation.

2. It would be even better if there were a more detailed explanation—such as a theoretical analysis—of why SSL mitigates dimensional collapse. Currently, the paper demonstrates the effectiveness of SSL through experimental results, but the underlying reasons for its success lack a more compelling, supportive explanation.

**Other Comments Or Suggestions:**

Please see above strengths and weaknesses

**Other Strengths And Weaknesses:**

+ This paper explores the extreme UniDA scenario, which is more challenging and rigorous compared to previous UDA approaches. This research problem is more reflective of real-world situations and holds significant practical importance.

+ The observed phenomenon of dimensional collapse is very intriguing and has the potential to inspire further research in the field.

- The primary contribution of this paper is addressing the issue of dimensional collapse using SSL methods. Since SSL has been widely applied in domain adaptation, we suggest that the authors dedicate a section to thoroughly discuss this issue, in order to better emphasize the novelty of their approach.

**Questions For Authors:**

No additional questions.

**Relation To Broader Scientific Literature:**

It can be applied to addressed dimension collapse problem in many fields.

**Theoretical Claims:**

No theoretical claims

---

> ### Author Rebuttal · Authors · 2025-04-01
>
> We thank the reviewer for the constructive feedback and are glad to hear that our research problem resonates with real-world scenarios and that the observed dimensional collapse is found to be intriguing. The reviewer’s main concerns lie in the novelty of using SSL and the lack of experiments on partial domain adaptation (PDA) and open set domain adaptation (OSDA). We address these concerns below.
>
> **Novelty of using SSL to address the dimensional collapse issue.**
> > There are concerns regarding the novelty of using SSL to address the dimensional collapse issue. The authors should further clarify this approach and compare it with previous SSL methods used in domain adaptation.
>
> We refer the reviewer to the "Missing References" section of Reviewer tc8f, where we discuss the distinction between our work and the literature on self-supervised learning for domain adaptation.
>
>
> **Why SSL mitigate DC?**
> > The claims and evidence in general are clear and convincing. But why SSL works in addressing dimension collapse may need more explanation.
>
> Existing insights from the SSL literature offer useful intuitions. Contrastive learning frameworks (e.g., AlignUniform, SimCLR) demonstrate that negative samples and the uniformity loss serve as repulsive forces to the alignment loss, helping to prevent dimensional collapse. On the other hand, non-constrastive methods like SimSiam and Barlow Twins show that using asymmetric architectures and encouraging feature decorrelation can promote diverse representations across dimensions, thereby mitigating collapse. We will incorporate these insights into the latest version of our paper.
>
> **Experiments on PDA and OSDA**
>
> Thank you for the suggestions! We have now included the results for PDA and OSDA below.
>
> * PDA (21/10/0, Office31)
>
> |   | A2D | A2W | D2A | D2W | W2A | W2D | Avg |
> | -------- | -------- | -------- | -------- | -------- | -------- | -------- | -------- |
> |Source-Only *| 90.4| 79.3 | 79.3 | 95.9 | 84.3 | 98.1 | 87.8
> | CMU*     | 84.1 | 84.2 | 69.2 | 97.2 | 66.8 | 98.8 | 83.4
> | DANCE* |  77.1 | 71.2 | 83.7 | 94.6 | 92.6 | 96.8 | 86.0
> | UAN | 82.3| 83.1 | 69.8 |  96.6 | 64.2 | 98.4 | 82.4
> | UAN + SSL | 91.1 | 84.6 | 81.2 | 97.8 | 83.6 | 98.6 | 89.5(+7.1) |
>
> *: from "LEAD: Learning Decomposition for Source-free Universal Domain Adaptation" CVPR'24
>
> These results indicate that UAN+SSL significantly outperforms previous methods in PDA settings. Interestingly enough, the scenario corresponds to $\pi_s=0.68$, which is close to the extreme UniDA setting, and all prior domain-matching methods underperform the SO approach, reinforcing our observation.
>
> * OSDA (0/10/11, Office31)
>
> |   | A2D | A2W | D2A | D2W | W2A | W2D | Avg |
> | -------- | -------- | -------- | -------- | -------- | -------- | -------- | -------- |
> | CMU*     | 70.5     | 71.6     | 81.2 | 80.2 | 70.8 | 70.8 | 74.2 |
> | DANCE* | 74.7 | 82.0 | 82.1 | 68.0 |  82.5 | 52.2 | 73.6 |
> | UAN | 67.8 | 68.4 | 78.8 | 75.4 | 66.3 | 68.2 | 70.8
> | UAN + SSL | 68.1 | 69.2 | 79.3 | 76.1 | 67.4 | 69.3 | 71.6 (+0.8)
>
> *: from "Subsidiary Prototype Alignment for Universal Domain Adaptation" NeurIPS'22
>
>
> The result suggest that incorporating SSL does not significantly affect performance. In the setting where $\pi_s=0$, the critical challenge lies in identifying target-private classes rather than domain matching, which may explain why SSL does not substantially influence the performance.
>
>
> **Ablaion study is only conducted on a subtask of the small Office31 dataset**
>
>
> We have included a more comprehensive ablation study on the entire Office-31 dataset, which shows that alignment alone provides limited benefit, uniformity leads to a slight performance improvement, and combining both yields the best results.
>
>
> |   | A2D | A2W | D2A | D2W | W2A | W2D | Avg |
> | -------- | -------- | -------- | -------- | -------- | -------- | -------- | -------- |
> | $\mathcal{L}_s$     |   70.1   |     64.5 | 62.5 | 68.4 | 59.7 | 63.1 | 64.7 |
> | $\mathcal{L}_s + \mathcal{L}\_{\text{align}}$    |    71.5  | 64.6     | 60.2 | 67.8 | 59.1 | 64.2 | 64.6 (-0.1)
> | $\mathcal{L}_s  + \mathcal{L}\_{\text{uniform}}$     | 72.2     | 65.1     |63.3 | 68.8 | 60.8 | 64.7 | 65.8 (+1.1)
> | $\mathcal{L}_s  + \mathcal{L}\_{\text{align}}  + \mathcal{L}\_{\text{uniform}}$       | 72.0     | 67.2     |64.7 | 70.1 | 62.1 | 64.9 | 66.8 (+2.1)

---

### Official Review · Reviewer_yX6d · 2025-03-12

**Overall Recommendation:** 3

**Summary:**

This paper addresses the universal domain adaptation problem, which is useful in real-world applications. They identify the failure of partial domain matching by dimensional collapse and propose to jointly leverage the alignment and uniformity techniques to avoid dimensional collapse. Experiments on four datasets and benchmarks highlight its superior performance.

**Claims And Evidence:**

Yes

**Essential References Not Discussed:**

No

**Experimental Designs Or Analyses:**

The experimental designs miss important PDA and OSDA setups.

**Methods And Evaluation Criteria:**

Traditional evaluation criteria [1,2] in univeral domain adaptation always include evaluations under partial-set domain adaptation (PDA), open-set domain adaptation (OSDA), and open-partial-set domain adaptation (OPDA) settings. However, the author only evaluate under last setting without PDA and OSDA.

[1] Universal domain adaptation through self supervision, NeurIPS 2020

[2] LEAD: Learning Decomposition for Source-free Universal Domain Adaptation, CVPR 2024

**Other Comments Or Suggestions:**

How about replace SSL with other pretext tasks, such as jigsaw puzzles and rotation?

**Other Strengths And Weaknesses:**

Strengths
1. The paper addresses the univeral domain adaptation problem, which is a challenging and practical scenario.
2. The paper is well written and easy to follow.
3. The paper provides extensive experiments, showing the effectiveness and versatility of the proposed method.

Major Weaknesses
1. The authors only use CNN backbones. More ablations on ViT backbone should be added, as it demonstrates strong generalization and adaptation performances compared with CNNs.
2. Lack of theoretical insights in support of the proposed method.
3. Comparison with more recent methods [1,2] should be included.
4. The novelty is limited. The self-supervised learning and uniformity loss have been widely used in universal domain adaptation.

[1] LEAD: Learning Decomposition for Source-free Universal Domain Adaptation, CVPR 2024

[2] Universal domain adaptation via compressive attention matching, ICCV 2023

**Questions For Authors:**

What is the performances under PDA and OSDA setups?

**Relation To Broader Scientific Literature:**

The proposed framework could be potentially helpful to other literature.

**Theoretical Claims:**

No Theoretical Claims

---

> ### Author Rebuttal · Authors · 2025-04-01
>
> Thank you for the constructive feedback. We address the raised concerns below.
>
> **Novelty and contributions**
>
> While SSL has been explored in various DA contexts, we respectfully argue that our contribution lies beyond using SSL for UniDA.
>
> Prior SSL for UniDA works typically rely on pretext tasks (e.g., rotation, jigsaw), which are ineffective in mitigating DC and fail under extreme UniDA settings, as shown in our comparison table below (**Comparison with other SSL pretext tasks**). In contrast, we use both contrastive learning and non-contrastive learning methods, which are explicitly designed to preserve representation diversity and combat dimensional collapse (DC) [1]. To the best of our knowledge, we are the first to apply these techniques in UniDA specifically to address DC, making our use of SSL novel in both motivation and application.
>
> We also highlight that **identifying the previously overlooked challenge of extreme UniDA is itself a key contribution**, as it draws attention to the limitations of current benchmarks and provides a foundation for systematically addressing the problem through improving representation quality.
>
> **Ablations on ViT backbone**
>
> We initially excluded this analysis because it is not commonly included in UniDA literature, even in recent works (e.g., LEAD, MLNet), except for [2], which specifically investigates ViT architectures. The following tables present the performance of ViT on both extreme UniDA and general UniDA settings for DomainNet. The results follow a similar trend to those observed with CNNs: improvements are more pronounced in the Extreme UniDA setting compared to the standard UniDA case.
>
> * Extreme UniDA
>     |  | P2S | P2R  | S2P | S2R | R2P | R2S | Avg
>     | - | - | - | - | - | - | - | - |
>     | UAN     | 44.69     | 59.18     | 40.96  | 57.44 | 39.70 | 36.07 | 46.34
>     | UAN + SSL     | 51.57     | 61.12     | 48.24 |  58.22 | 42.95  | 37.49 | 49.93 (+3.59)
>
> * General UniDA
>     |  | P2S | P2R  | S2P | S2R | R2P | R2S | Avg |
>     | - | - | - | - | - | - | - | - |
>     | UAN     | 44.38     | 62.58     | 34.11 | 53.31 | 51.34  | 41.62 | 47.89
>     | UAN + SSL     | 46.25     | 62.81     | 38.74 | 54.76 | 53.26 | 42.11 | 49.65 (+1.76)
>
> **Lack of theoretical insights in support of the proposed method**
>
> We agree that understanding why SSL mitigates DC in extreme UniDA is a valuable theoretical pursuit. While seminal SSL methods (e.g., SimCLR, SimSiam, Barlow Twins) were initially driven by intuition and empirical success, later works sought theoretical explanations [3, 4]. We believe that our philosophical insights and empirical results lay a strong foundation for future theoretical study.
>
> **Comparison with more recent methods**
>
> Thank you for highlighting these baselines. We were aware of them but initially excluded them due to differing settings. LEAD uses a source-free setup without target data during training; we now include its results on DomainNet, where it underperforms compared to several baselines under the extreme UniDA setting. [2] uses a ViT backbone, making it incompatible with our CNN-based setup. It was also excluded in the MLNet (AAAI 2024) paper, and its code is unavailable, preventing reproduction within our limited timeframe.
>
> |  | P2R | R2P | P2S| S2P | R2S| S2R| Avg|
> | - | - | - | - | - | - | - | - |
> |CMU | 30.1 | 42.4  | 34.1 | 24.3  |32.2 | 34.1 | 32.8 |
> | UniOT     |  38.1 | 29.8 | 30.8 | 29.3 | 29.1|  38.3 | 32.6
> | LEAD     |   17.3| 16.5 | 15.4 | 14.8| 15.8 | 15.3 | 15.9 |
>
> **Comparison with other SSL pretext tasks**
>
> We compare three different approaches to SSL in DA:
>
> 1. pretext tasks: Minimize domain gap via auxiliary tasks like jigsaw puzzles or rotation prediction (Bucci et al., Xu et al.).
> 2. prototype alignment: Align target samples to source/target prototypes using entropy minimization (DANCE).
> 3. Contrastive & non-contrastive: The focus of our paper—these methods promote representation diversity to avoid collapse, using contrastive losses (AlignUniform) or asymmetric architectures without negatives (SimSiam).
>
> |  | H-score on Office31|
> | - | - |
> | Rotation/Location (Xu et al.)  |   59.7  |
> | Jigsaw Puzzles (Bucci et al.)  |    63.4  |
> | DANCE  |   61.2   |
> | AlignUniform  |  71.8    |
> | SimSiam   |    72.3  |
>
> This suggests that contrastive and non-contrastive methods, which explicitly tackle DC, are effective in extreme UniDA—unlike previously explored SSL approaches, which struggle in this setting.
>
> **Results on PDA and OSDA**
>
> We refer the reviewer to the "Results on PDA and OSDA" section of Reviewer sBWy.
>
> [1] “Rethinking The Uniformity Metric in Self-Supervised Learning”, ICLR 2024
>
> [2] “Universal Domain Adaptation via Compressive Attention Matching”, ICCV 2023
>
> [3] “Understanding Dimensional Collapse in Contrastive Self-supervised Learning”, ICLR 2022
>
> [4] “How Does SimSiam Avoid Collapse Without Negative Samples? A Unified Understanding with Self-supervised Contrastive Learning”, ICLR 2022

---

> > ### Comment · Reviewer_yX6d · 2025-04-08
> >
> > I want to thank the authors for the rebuttal, most of my concerns are addressed and I therefore increase the score to 3.

---

> > > ### Author Response · Authors · 2025-04-09
> > >
> > > We thank the reviewer for carefully considering our rebuttal and for increasing the score. We're glad that our response addressed your concerns.

---

### Official Review · Reviewer_UDYW · 2025-03-15

**Overall Recommendation:** 3

**Summary:**

The paper investigates the cause of the performance degradation of partial domain matching (PDM) in (extreme) universal domain adaptation (UniDA). Specifically, the paper presents the failure mode of PDM resulting from dimensional collapse (DC) in target representations, in extreme UniDA where the source-private classes are abundant. To address this issue, the paper proposes to incorporate self-supervised learning on unlabeled target data during training. Ablation studies and empirical evaluations are presented.

---

### Post-rebuttal

I find the clarification in authors' rebuttal helpful and encourage authors to incorporate it in manuscript. I have increased my score.

**Claims And Evidence:**

The primary claim of the paper is that DC results in the failure of PDM in extreme UniDA settings. The evidence comes from the analyses on different applications of SSL loss functions, employing the Wang and Isola (2020) framework.

**Essential References Not Discussed:**

There are no significant missing references (with the caveat that the setting is sensible, more in Other Strengths and Weakness part).

**Experimental Designs Or Analyses:**

The illustrative experiments on the influence of extreme UniDA on PDM, in terms of DC, are based on singular values (consistent with previous approaches), as well as the relation of DC and loss functions (employing particularly the framework of Wang and Isola, 2020).

The experiments (Section 5) are designed to showcase the potential benefit of SSL for PDM, and the generalization across different DA settings.

**Methods And Evaluation Criteria:**

The method starts from analyzing SSL loss functions to demonstrate the pitfalls of PDM in UniDA, and followed by incorporating unlabeled target data into training, in addition to source-labeled data.

The evaluation criteria depend on the data, and include accuracy, H-score (based on accuracy), and various visualizations.

**Other Comments Or Suggestions:**

(additional comment on material organization)

In the Introduction, the settings and approaches for UniDA and UDA are introduced interchangeably. Considering the fact that the abbreviations are very similar, it might worth considering rearranging the material to make the content more consistent locally (e.g., no jumping back and forth between UDA and UniDA, now that the object of interest is UniDA).

**Other Strengths And Weaknesses:**

I have one particular concern: the utilization of target data (although unlabeled) makes the setting no longer extreme UniDA.

On the one hand, if the distinction of settings (e.g., UDA or UniDA) is w.r.t. both the label sets (explicitly) and constraint on the corresponding features (implicitly), then the gap "Extreme UniDA" in the spectrum of UniDA (Fig. 1) is important and meaningful to address. However, the introduction of target data without labels during training shifts the setting to a learning problem instead of a DA problem. The previous approaches compared are DA methods, and therefore, are not sufficient to demonstrate the benefits of the proposed approach.

On the other hand, if the distinction of settings is only w.r.t. the labels themselves, then the setting of "Extreme UniDA" becomes an ill-posed problem, since there is no additional assumption/leverage on how target features (not labels) relate to the source domain. According to the paper, the appearance of unlabeled data (from target domain) is still within Extreme UniDA.

**Questions For Authors:**

Can authors share further clarifications/discussions w.r.t. the concern about the shift in setting due to the utilization of target data (even if unlabeled) during training?

**Relation To Broader Scientific Literature:**

The paper is related to different settings/instantiations of domain adaptations.

**Theoretical Claims:**

The evidence of the paper primarily comes from empirical evaluations and illustrations.

---

> ### Author Rebuttal · Authors · 2025-04-01
>
> We would like to thank the reviewer for the constructive feedback. The primary concern raised is whether the utilization of unlabeled target data during training is still consistent with the Extreme UniDA setting. We address this concern below:
>
> **Concern about the utilization of target data, which makes the setting no longer extreme UniDA.**
>
> We appreciate the reviewer’s perspective and acknowledge that different lines of work may interpret or label these setups differently. However, our approach remains squarely within the unsupervised domain adaptation (UDA) paradigm—specifically, universal domain adaptation (UniDA). In this framework, both label sets (label shift) and underlying data distributions (covariate shift) can differ between source and target domains—aligning with the first category the reviewer mentioned—and having access to unlabeled target data during training is a standard assumption.
>
> We will clarify in the revised manuscript that having unlabeled target data during training is not an additional assumption outside of domain adaptation; rather, it is intrinsic to UDA and UniDA. Removing all target data during training would place the problem under settings like domain generalization [1], zero-shot transfer [2], or test-time adaptation [3], which focus on generalizing without any prior exposure to the target domain. In contrast, both UDA and UniDA explicitly require access to unlabeled target data, and every baseline PDM method we compare (e.g., UAN, CMU, UniOT) operates under this same premise.
>
> Therefore, while we recognize that terminological nuances may vary, our work consistently follows the established UniDA framework. We do not shift toward a purely supervised or semi-supervised learning problem on the target domain; instead, we remain entirely within the standard assumptions of UniDA by incorporating unlabeled target data during training.
>
> [1] Li et al., “Learning to generalize: Meta-learning for domain generalization”, AAAI 2018
>
> [2] Radford et al., “Learning Transferable Visual Models From Natural Language Supervision”, ICML 2021
>
> [3] Wang et al., “Tent: Fully test-time adaptation by entropy minimization”, ICLR 2021
>
>
> **Interchangeable usage of UniDA and UDA**
>
> Thank you for your helpful suggestion. We agree that the interchangeable use of UniDA and UDA may cause confusion. While our primary focus is on Universal Domain Adaptation (UniDA), we include key references from the Unsupervised Domain Adaptation (UDA) literature to provide necessary context. In the revision, we will clarify this distinction and streamline the discussion of UDA works to maintain focus.

---

### Official Review · Reviewer_tc8f · 2025-03-18

**Overall Recommendation:** 4

**Summary:**

This work focuses on the problem of extreme Universal Domain Adaptation (UniDA). Firstly, UniDA considers a domain adaptation problem where a model has to be trained with a labeled source domain and an unlabeled target domain, such that the label sets of source and target domains are disjoint (i.e. some classes are source-private, some are shared, and some are target-private). The extreme UniDA problem considers the cases of having a very large number of source-private classes, i.e. many source classes are absent from the target data. This paper analyzes the partial domain matching (PDM) approaches for UniDA and finds that they fail on extreme UniDA due to dimensional collapse (DC). To address DC, they propose to use an existing SSL method that encourages learned representations to be uniformly distributed on the unit hypersphere, in order to preserve their intrinsic structure. Finally, they perform experiments on extreme UniDA and show that existing approaches can be improved using the SSL method.

**Claims And Evidence:**

* Fig. 2: It would be good to report common classes accuracy and target-private accuracy separately (apart from the H-score), so we have a clear picture of whether both of these are worse than SO for high $\pi_s$ or if one or the other is worse. This could also give additional insights into which part of the UniDA algorithm needs to be improved for extreme UniDA.

* Fig. 4 (b, d): The analysis is interesting and informative, but it uses only older works (2022 and before). It would be stronger if the same is evaluated with newer works/methods (like LEAD, Qu et al. CVPR 2024 or another work from 2023-2025).

**Essential References Not Discussed:**

* SSL-based UniDA work (Kundu et al. 2022) was only mentioned in experimental settings in Appendix D.3. However, this work designed a self-supervised pretext task specifically for Universal DA and should be discussed and compared with the proposed method. This is also currently missing from Table 1 (comparison of existing SSL approaches for UniDA).

* Another SSL-based UniDA work [W1] propose a self-supervised adaptive memory network with consistency regularization, and should be discussed and compared with the proposed method.

* A highly relevant SSL-based DA work [W2] was not discussed and cited in the paper. It explores the use of existing pretext tasks for DA, highlights their limitations, and proposes a new pretext task specifically designed for closed-set DA. It should be discussed in the related work.

[W1] Zhu et al., “Self-supervised Universal Domain Adaptation with Adaptive Memory Separation”, ICDM 2021

[W2] Kundu et al., “Concurrent Subsidiary Supervision for Unsupervised Source-Free Domain Adaptation”, ECCV 2022

**Experimental Designs Or Analyses:**

* Overall experimental design seems reasonable and valid. And the analysis experiments added to motivate different ideas in Sec. 3 and 4 make the paper very interesting to read.

* Sec. 4.2: Is the self-supervised loss same as the uniformity loss from Sec. 4.1? If yes, it would be good to use the same notation throughout so that it's easier to follow. If not, then better clarify what self-supervised loss is and how it is different from uniformity loss.

**Methods And Evaluation Criteria:**

Yes, the paper uses standard evaluation criteria similar to prior UniDA works and develops new analysis experiments that also seem to be valid and useful.

**Other Comments Or Suggestions:**

None

**Other Strengths And Weaknesses:**

* The writing of this paper is solid and the analysis experiments in Sec. 3 and 4 strongly motivate the proposed approach.

* A minor weakness is that the SSL method is not novel and re-purposed from Wang and Isola (2020). However, this work seems to be the first to use it for UniDA.

**Questions For Authors:**

Please address the concerns listed above. Overall, the paper is well-written and well-motivated with good results. However, I have some concerns regarding essential references not being discussed properly in the paper, apart from other minor concerns in the experiments. Hence, my rating is currently “weak accept” but I am willing to update my rating based on the rebuttal.

## Update after rebuttal

I thank the authors for their efforts in the rebuttal. Since my major concerns are resolved, I upgrade my rating to "accept".

**Relation To Broader Scientific Literature:**

* This paper analyzes the failure of existing UniDA methods for the difficult setting of extreme UniDA, which is novel and interesting.
    * Specifically, they analyze the partial domain matching approaches and find that they fail due to dimensional collapse (DC).
    * To resolve the DC problem, they re-purpose an existing SSL method (hypersphere-based uniformity loss).
    * Finally, they show improved results for extreme UniDA by incorporating the SSL method into existing UniDA techniques like UAN, UniOT, and MLNet.

* While UniDA is a more complex and niche setting compared to general DA, it is a more practical setting. Further, the analysis experiments in this paper are quite valuable and interesting to motivate future work.

**Theoretical Claims:**

Not applicable

---

> ### Author Rebuttal · Authors · 2025-04-01
>
> We thank the reviewer for the constructive feedback and are glad our analysis strongly motivates the proposed approach. Below, we address the raised concerns.
>
>
> **Missing references**
>
> We appreciate the reviewer’s feedback on our discussion of SSL for DA. While prior work primarily employs SSL to minimize domain gap—either through pretext tasks or by aligning low-uncertainty samples with class prototypes [2, 3, 4]—our focus is on preventing dimensional collapse (DC) in extreme UniDA. This different goal leads us to adopt SSL paradigms specifically designed to counteract collapse by enhancing feature diversity. Our methods include both contrastive (AlignUniform) and non-contrastive (SimSiam, Barlow Twins) approaches that promote uniformity and decorrelation to mitigate DC [5], rather than simply using pretext tasks for domain matching. We also present performance comparisons on extreme UniDA in the table below (see "Comparison with other SSL pretext tasks" of reviewer yX6d), showing that pretext-based and prototype-alignment methods fall short of the performance achieved by approaches specifically targeting DC.
>
>
> The above content will be included in Section 4 for a more comprehensive comparison in our latest version. We now discuss how our approach differs from the specific works [1, 3, 4].
>
> Kundu et al. [1] propose "sticker-intervention," a pretext task that improves domain preservation over traditional SSL tasks. We will include it in the related work.
>
> Zhu et al. [3] propose a framework that is highly similar to DANCE [2], comprising prototype alignment (referred to as adaptive memory in [3]) and an entropy separation module. As [3]'s code is unavailable, we use DANCE as a proxy given their similar design. DANCE struggles in extreme UniDA (see Table 3–5), making it a reasonable reference point. We will include a discussion of [3] in the related work section.
>
> SPA [4] introduces an add-on module that performs adaptation at the mid-level layers, as these layers are shown to exhibit lower negative transfer. Their method leverages a Bag-of-Words-inspired pretext task to learn distinct visual word prototypes and promotes prototype alignment via entropy minimization. Their work mainly improves domain matching, which is orthogonal to our work that tackle DC. As their module is designed to be an add-on, it can serve as a complementary component to our method. Since the code is not publicly available, we currently include their method only in the related work discussion and will add it as a baseline once the code is released.
>
> [1] “Concurrent Subsidiary Supervision for Unsupervised Source-Free Domain Adaptation”, ECCV 2022
>
> [2] “Universal Domain Adaptation through Self-Supervision”, NeurIPS 2022
>
> [3] “Self-supervised Universal Domain Adaptation with Adaptive Memory Separation”, ICDM 2021
>
> [4] “Subsidiary Prototype Alignment for Universal Domain Adaptation”, NeurIPS 2022
>
> [5] “Rethinking The Uniformity Metric in Self-Supervised Learning”, ICLR 2024
>
> **Report common-class accuracy and target-private accuracy separtely in Figure 2.**
>
> Thank you for the suggestion! Due to character limitations, we were unable to include the full tables here, but we summarize the observed trends below:
>
> * H-score: The H-score decreases as $\pi_s$ increases, as shown in Figure 2. Additionally, UniOT and UAN perform worse than SO under high $\pi_s$.
> * Common-class accuracy: This also decreases with increasing $\pi_s$. UniOT performs slightly worse than SO at high $\pi_s$, while UAN shows a clearly lower performance than both.
> * Target-private accuracy: Accuracy on target-private classes declines as $\pi_s$ increases for all methods. Both UniOT and UAN yield slightly lower scores than SO.
>
>
>
>
> **While the analysis in Figure 4 is interesting and informative, it would be more complete with the inclusion of more recent methods.**
>
> We agree that more recent methods should be included. Since LEAD is a source-free method and does not use a domain-matching loss, we have instead included MLNet (AAAI'24), which employs mutual nearest neighbors (MNN) for domain matching. This approach is conceptually similar to our distance-based baseline, but its mutual filtering enforces a stricter alignment.
>
>
>
> * Avg. error rate of importance weight function during training
>
>    | |UAN | CMU | Energy | Distance | MNN |
>    | -   | - | - | - | - | - |
>    | $\pi_s=0.25$ | 0.29| 0.30     | 0.33     | 0.26 | 0.24 |
>    | $\pi_s=0.75$ | 0.59| 0.53     | 0.51     | 0.49| 0.41 |
>
>     The MNN baseline shows a significant improvement. However, it is still far from the threshold (0.15–0.2), as shown in Figure 4 (a), required to outperform the Source-Only baseline.
>
> **Is the self-supervised loss from Sec. 4.2 the same as the uniformity loss from Sec. 4.1?**
>
> The self-supervised loss refers to alignment + uniformity loss, where alignment encourages class-wise aggregation and uniformity prevents DC. We’ll clarify this in the revision.

---

### Decision · Program_Chairs · 2025-05-01

**Decision:**

Accept (poster)

**Comment:**

There are two major concerns. One is regarding some experimental setup and results, and the other is regarding the novelty of the proposed method compared with existing SSL-based domain adaptation methods. During rebuttal, the empirical concern has been addressed, while the technical novelty issue is not addressed (the authors only mentioned they *would* discuss the distinction between their work and existing SSL methods for domain adaptation). This makes this paper a borderline paper.